# Gibberellins and Heterosis in Crops and Trees: An Integrative Review and Preliminary Study with *Brassica*

**DOI:** 10.3390/plants9020139

**Published:** 2020-01-22

**Authors:** Karen P. Zanewich, Stewart B. Rood

**Affiliations:** Department of Biological Sciences, University of Lethbridge, Lethbridge, Alberta, T1K 3M4, Canada; zanewich@uleth.ca

**Keywords:** canola, hybrid vigor, maize, phytohormones, poplar, root growth, sorghum

## Abstract

Heterosis, or hybrid vigor, has contributed substantially to genetic improvements in crops and trees and its physiological basis involves multiple processes. Four associations with the phytohormone gibberellin (GA) indicate its involvement in the regulation of heterosis for shoot growth in maize, sorghum, wheat, rice, tomato and poplar. (1) Inbreds somewhat resemble GA-deficient dwarfs and are often highly responsive to exogenous GA_3_. (2) Levels of endogenous GAs, including the bioeffector GA_1_, its precursors GA_19_ and GA_20_, and/or its metabolite GA_8_, are higher in some fast-growing hybrids than parental genotypes. (3) Oxidative metabolism of applied [^3^H]GAs is more rapid in vigorous hybrids than inbreds, and (4) heterotic hybrids have displayed increased expression of GA biosynthetic genes including *GA 20-oxidase* and *GA 3-oxidase*. We further investigated *Brassica*
*rapa*, an oilseed rape, by comparing two inbreds (AO533 and AO539) and their F_1_ hybrid. Seedling emergence was faster in the hybrid and potence ratios indicated dominance for increased leaf number, area and mass, and stem mass. Overdominance (heterosis) was displayed for root mass, leading to slight heterosis for total plant mass. Stem contents of GA_19,20,1_ were similar across the *Brassica* genotypes and increased prior to bolting; elongation was correlated with endogenous GA but heterosis for shoot growth was modest. The collective studies support a physiological role for GAs in the regulation of heterosis for shoot growth in crops and trees, and the *Brassica* study encourages further investigation of heterosis for root growth.

## 1. Introduction

### 1.1. Gibberellins and Heterosis in Crops and Trees: Four Lines of Evidence

Heterosis, or hybrid vigor, represents hybrid performance that exceeds that of both parents [1]. Heterosis is common in crop plants and in some trees and has been responsible for a substantial proportion of the genetic improvement through plant breeding over the past century [2,3]. However, while heterosis is widely utilized for agriculture, horticulture and silviculture, and despite a century of investigation, its physiological basis is poorly understood [4].

Studies indicate that there are multiple contributing genetic and physiological factors [5,6,7] and phytohormones should be involved since these provide master regulators of plant form and function [8,9]. Of the phytohormones, there is substantial evidence that the promotive regulator gibberellin (GA) is especially involved in heterosis for shoot growth. There have been preliminary reviews of this association [10,11] and we extend the analyses to incorporate recent findings, expand the range of study approaches and increase the number of plant species.

### 1.2. Inbreds are Highly Responsive to Exogenous GA_3_

With plant breeding and genetics programs, several single gene dwarf mutants were identified in maize (or corn, *Zea mays* L.), peas, *Arabidopsis*, *Brassica* and other plants [9]. Some dwarfs have mutations in GA biosynthetic genes and can be restored through the exogenous application of gibberellic acid (GA_3_), indicating that their small size is due to a GA deficiency. With inbreeding, shoot growth is often depressed and with short stature, inbreds somewhat resemble dwarf mutants. A half century ago, Nickerson [12] recognized this similarity and applied GA_3_ to different maize genotypes. He found that maize inbreds were highly responsive to exogenous GA_3_ suggesting that, like the dwarf mutants, the inbreds could be limited by low levels of endogenous GAs. At that same time, Chawdhry [13] found variation in responsivity to GA_3_ across tomato genotypes and proposed a link between GAs and heterosis. The association was emphasized by Sarkissian et al. [14] and supported in an early review by Paleg [8]. Further studies with maize, the model plant for heterosis (Table 1), and other plants (Table 2) have confirmed the common correspondence, with smaller genotypes including inbreds often being highly responsive to exogenous GA_3_.

### 1.3. Hybrids Have Higher Levels of Endogenous GAs

The hypothesis that inbreds are GA deficient was subsequently tested with advancing techniques including selective GA bioassays, GA immunoassays, and definitive physicochemical analyses with combined gas chromatography-mass spectrometry (GC-MS). For GC-MS analyses, internal standard [^2^H]GAs are added to account for variable losses during the extensive purification [15,16]. Studies with different species and in different research labs supported the hypothesis, as heterotic hybrids contain higher levels of endogenous GAs (Table 1 and Table 2). This provides the second line of evidence supporting a regulatory role of GAs in heterosis for shoot growth.

Multiple GAs are native in plants and GA_1_ is probably the bioeffector GA in most angiosperms for shoot elongation and some other processes (Figure 1a). The biosynthetic pathways have been determined and the “early 13-hydroxylation pathway” is probably the primary pathway in most crops and broad-leaved trees (Figure 1b). Another biosynthetic pathway leads to GA_4_, a bioactive GA [42] or an alternate GA_1_ precursor (Figure 1b) [43]. With 2-hydroxylation, GA_1_ is inactivated to form GA_8_ and different GAs are often analyzed to provide a record of present (GA_1_), past (GA_8_) and future (GA_19_, GA_20_, GA_4_) GA biosynthesis (Figure 1b). Researchers have primarily analyzed these endogenous GAs which may display different patterns across the genotypes (Table 1 and Table 2).

### 1.4. GA Metabolism is Rapid in Heterotic Hybrids

Analyses of endogenous GAs provide instantaneous measurements, while studies of GA metabolism capture physiological effects over hours or days. [^3^H]GA_20_ has been applied to parents and hybrids and the [^3^H] products analyzed. [^2^H]GA_20_ was also applied, permitting GC-MS confirmation of the 3-hydroxylation to [^2^H]GA_1_ (Table 1 and Table 2; Figure 1b). In maize and sorghum, metabolism of [^3^H]GA_20_ was faster in heterotic hybrids than parental genotypes, as was the 2-hydroxylation of [^3^H]GA_1_ to [^2^H]GA_8_ or its glucosyl conjugates. These findings indicate faster oxidative metabolism in fast-growing hybrids and provide the third line of evidence for a role of GAs in the regulation of heterosis for shoot growth.

### 1.5. Expression of GA Biosynthetic Genes is Increased in Some Hybrids

After the GA biosynthetic sequence was determined, the responsible genes were identified and cloned [42,44]. The multi-gene *GA 20-oxidase* family is responsible for sequential conversions from GA_53_ through GA_44_ and GA_19_ to GA_20_ (Figure 1b). GA_19_ is often an abundant endogenous GA and its conversion to GA_20_ is regarded as rate-limiting and hence, regulatory [44]. This gene has been especially studied and its expression has been associated with heterosis in some plants (Table 1 and Table 2). The final conversion from GA_20_ to GA_1_ is enabled by *GA 3-oxidase*, another multi-gene family (formerly *3β-hydroxylase*), and the expression of this gene has also been found to be correlated with heterosis (Table 2). This association between the expression of the GA biosynthetic genes and hybrid vigor provides the fourth line of evidence supporting the role of GAs in the regulation of heterosis for shoot growth in crops and trees.

With multiple studies over a half century, by different researchers and with different plants, there has been substantial support for the association between GAs and heterosis for shoot growth. However, there has also been opposing evidence. In some studies, there have been no correspondences but there are confounding influences. A challenge is the tissue sampling since GAs are differentially distributed within plants, commonly with higher concentration in the shoot apices and subtending elongation zones [45]. Below those zones, the concentrations are lower and if whole stems are sampled there could be greater dilution by the larger stems of hybrids, thus obscuring and even inverting the endogenous GA pattern. This challenge can be increased for trees, which are much larger, and consequently bark scrapings are often sampled (Table 2). However, the locations relative to the growth zones remain a complexity and the timing of sampling further challenges these studies [36,45].

Another complexity relates to the type of growth. The association between GA and stem height is well established across plants, and there are also correspondences with other elongation measures such as leaf lengths, but there may be weaker associations with some other growth measures. A study by Auger et al. [22] applied an interesting approach with GA dwarfing genes in maize, and heterosis persisted. Auger et al. [22] considered this as opposing a regulatory role for GAs but the dwarfing genes would lead to GA reduction [46] rather than elimination, which would be lethal. Consequently, without analyses of the GA contents, that study was less complete.

## 2. Results

### 2.1. Gibberellins and Heterosis in Brassica

To further explore the growth components that contribute to heterosis, and to investigate their association with endogenous GAs, *Brassica* was used as another study system. Heterosis is observed in some *Brassica* hybrids [47,48] and genetic diversity is incorporated with the allotetraploid genotypes of some rapeseed and canola cultivars [49]. *Brassica* is a close crop relative to *Arabidopsis*, allowing for the application of knowledge from that primary model plant system, which also displays heterosis [40,50,51]. Consequently, this study was undertaken with a triplet family consisting of two inbreds, AO533 and AO539, and their F_1_ hybrid, and we expected: (1) heterosis in the hybrid and (2) correspondence between GA content and stem growth within and across the genotypes.

### 2.2. Growth and Dominance Patterns

Initial seedling emergence was more rapid in the *Brassica* hybrid (Table 3). Subsequently, the parent AO539 and the F_1_ displayed similar seedling emergence percentages, while AO533 displayed slower and less complete emergence. This pattern of dominance or slight overdominance for the increased trait was also observed for most of the subsequent shoot characteristics; the F_1_ was slightly superior to AO539, while AO533 was consistently less vigorous (Table 3 and Table 4). This was the case for leaf number while leaf sizes were similar. As the product of these two traits, total leaf areas and leaf masses were greater in the F_1_ and AO539 than AO533 (Table 3 and Table 4, Figure 2).

The sequential harvests displayed the increasing growth of the *Brassica* plants, with the F_1_ displaying similar or slightly greater growth than AO539, and both showing greater growth than AO533 (Figure 2). There was apparently slight heterosis for stem mass and the largest difference was for root mass (Figure 2). The F_1_ root mass was consistently larger and there was a lag in root growth in AO533. This provided the greatest contribution to the total plant mass, which displayed slight heterosis (Figure 2).

The analyses of covariance included the combined results from harvests on days 21, 35 and 42, with day as the covariate. The primary factor of genotype provided significant effects (*p* < 0.05) for seven of the eight growth measures (Table 4). Subsequent post-hoc pairings with the marginal means confirmed the pattern displayed in the growth plots, with the F_1_ being similar to or slightly larger than AO539, while AO533 was smaller (Table 4, Figure 2).

Potence ratios (PR) quantified the degree of dominance with values substantially greater than 1 indicating overdominance, or heterosis (Table 4). PR averages from the three harvests are provided (Table 4) and were highest for juvenile height, which extended from the early advantage of the hybrid, and root mass, which contributed to the moderate overdominance for overall plant mass. Root sampling is imprecise due to some adhering substrate particles and the loss of some fine roots. The root rinsing with a water spray appeared complete and we expect similar root recoveries across the harvests and genotypes, providing a valid comparison. The challenge with root sampling should be recognized but the results indicate substantial heterosis for root growth with this triplet of genotypes.

### 2.3. Endogenous Gibberellins

Generally, similar patterns across the genotypes were observed for GA_1_ and its precursors, GA_20_ and GA_19_ (Figure 3). There were often lower GA contents at the early harvests and increases by day 42, when stem bolting was commencing. This was also during the transition to the reproductive phase (Figure 3) and it is likely that endogenous GA would be involved in that developmental process as well as stem elongation. There were no significant differences across the genotypes for GA content or concentration at day 21 or 35, and at day 42 GA contents per stem were significantly higher in the F_1_ hybrid than in either parent (Figure 3; ANOVA with combined GAs: *F*_(2,4)_ = 11.3, *p* = 0.023). Concentrations were higher in the F_1_ than in AO539 but not significantly different from AO533 (3.16, 2.60 and 3.83 ng/g for GA_19_; 2.50, 1.59 and 2.36 for GA_20_; and 2.65, 1.90, and 3.39 for GA_1_ in AO533, AO539 and the F_1_, respectively; ANOVA: *F*_(2,4)_ = 15.49, *p* = 0.013). These results support the involvement of endogenous GAs in the regulation of stem elongation but are uncertain relative to the relationship with heterosis for shoot growth.

## 3. Discussion

### 3.1. Heterosis and Gibberellins in Brassica

In this study, hybrid vigor, superior performance of the hybrid, was slight. As occurs for other plants [14,53,54], seedling emergence was faster in the hybrid. Subsequently, for the different growth traits, the hybrid performance matched or slightly exceeded the superior inbred parent. The expression of heterosis through the multiple traits supports a polygenic foundation and the hybrid would benefit from superior performance for many traits, both independent and coordinated [55]. We expected heterosis for shoot growth, but an unexpected finding was that heterosis was greatest for root growth. There have been fewer studies of heterosis and root growth but early research with tomato hybrids recognized heterosis for root growth [56] and this pattern has been subsequently demonstrated for sorghum and maize [53,57]. At the cellular level, heterosis for gene expression in roots has also been demonstrated, but the function of some genes is less clear [58].

Root mass is a product of root numbers and their sizes, including lengths. The observed overdominance for total root mass might reflect the additive benefit from dominance for higher root numbers and lengths, and it would be useful to resolve the root components and even cell sizes and numbers. If GAs were involved, this might relate to root lengths, since GAs contribute strongly to the regulation of elongation growth and GA manipulation through recombinant *GA 2-oxidase* resulted in denser but shorter root systems in hybrid poplar [59]. Heterosis for root growth might contribute to yield stability and could influence adaptation to drought or salinity [60].

Relative to endogenous GAs, GA_1_ and GA_19_ contents per stem were apparently higher in the hybrid on day 42, when the bolting stage commenced, and the hybrid displayed slight heterosis for increased stem mass. However, this *Brassica* study was preliminary and future research should focus on hybrid versus parent combinations that display greater hybrid vigor in the growth patterns [61,62], as has been useful for maize inbreds versus hybrids [15,17,23,63]. A comparison for *Brassica* could include family combinations with increased hybrid vigor [64] and it would be useful to analyze GA contents and especially GA_1_ along with its precursors and metabolite, as well as with the expression of the key GA biosynthesis genes [39,42] (Table 1). A genetic approach could also be undertaken such as investigating the influence on heterosis from dwarfing genes in *Brassica* that block GA biosynthesis or action [22,46,65]. Due to the genetic, molecular and physiological similarity, directions from studies of heterosis in *Arabidopsis* should also be productive and these have supported the involvement of phytohormonal regulation [50]. And while GA could provide a focus, there are likely to be coordinated interactions with other phytohormones and especially auxins [58,66], along with other signaling substances such as brassinosteroids [25].

### 3.2. A Final Consideration for GAs and Heterosis: Cause or Correlate

While this study with *Brassica* provided provisional outcomes regarding heterosis, it is clear that GAs are involved in the regulation of shoot growth in *Brassica*, as is observed for the other plants studied: maize, sorghum, wheat, rice and poplar (Table 1 and Table 2). There have been numerous studies that provide direct or indirect support for the role of GAs in the regulation of heterosis for shoot growth, including stem and leaf growth, in crops and trees. However, while the findings indicate that slow-growing inbreds are limited due to relative GA deficiency, or conversely that hybrids display increased GA biosynthesis and contents, the subsequent question arises, “how are the GA biosynthetic genes regulated?” Thus, GAs may serve as an intermediate between heterotic factors and especially genetic heterozygosity and the phenotypic outcome of hybrid vigor. Also, rather than individual genes being central, as is the case for single gene dwarf mutants of maize, *Brassica* and other plants, a number of heterozygous genes would probably contribute, providing a polygenic foundation. The consequent up-regulation of the GA biosynthesis genes would provide a mechanism to amplify the outcome, since this would promote a range of morphological and physiological traits that contribute to the phenotypic outcome of hybrid vigor.

## 4. Material and Methods

### 4.1. Plant Material and Growth Conditions

Seeds of two inbred parental lines of *Brassica rapa* L. (syn. *B. campestris* L.), AO533 and AO539, and their F_1_ hybrid, AO533 × AO539 (from the Agriculture Canada canola breeding program, Saskatoon, Canada), were sown in pots containing Metro-mix (Ajax, ON, Canada), a *Sphagnum* peat moss and vermiculite-containing potting medium. Plant pots were randomly arranged in the University of Lethbridge greenhouse (49°70′ N, 112°86′ W), with a 16 h photoperiod that was provided by a combination of natural sunlight and light from Na vapour lamps (260 µmol m^−2^s^−1^; Reflector PL90M [medium] N400; P.L. Light Systems Canada, Inc., Beamsville, ON, Canada), and with temperatures maintained between 21–24 °C. Pots were watered daily to saturation and fertilized monthly with a water-soluble 20-14-14 (N-P-K) fertilizer with supplemental trace elements (Professional Gardener Co., Ltd., Calgary, AB, Canada).

### 4.2. Growth Measurements

Seedling emergence was noted and height and developmental stages [52] were recorded biweekly. Ten individual plants of each genotype were harvested at each of five harvest dates, at 14, 21, 35, 42 and 70 days after planting and for each plant, leaf number, total leaf area using a portable LI-COR photometric area meter (Lincoln, NB, USA), and organ dry weights (leaf, stem and root) were measured. Prior to weighing, leaf and root tissues were oven dried, while stem tissues were flash frozen in liquid nitrogen and then freeze dried for dry weight determinations and to permit analysis of endogenous GAs.

### 4.3. Potence Ratios

Inheritance patterns were quantified by comparing the performance of the hybrids with that of the two parents [17,67]. The additive component of variation (H) was calculated as the absolute value of either parent (P_2_ represents the superior parent for that trait) minus the mean of the two parents, the midparent value (mp). The dominance component of variation (D) was calculated as the hybrid value minus mp and the potence ratios represent the overall degree of dominance as:Potence ratio (PR) = D/H = (Hybrid − mp)/(P_2_ − mp)

Recognizing some quantitative variation, PR values around 0 (from about −0.5 to 0.5) indicate additive inheritance with the hybrids intermediate between the parents. PR values from ~ −1.5 to −0.5 or 0.5 to 1.5 indicate dominance of the smaller or larger characteristic, respectively. PR values exceeding ~ 1.5 indicate overdominance or hybrid vigor, while values below ~ −1.5 indicate hybrid break-down.

### 4.4. Analyses of Endogenous Gibberellins

Gibberellins (GAs) were quantified in one of the two replicates, with each consisting of 5-6 stems from a genotype at a harvest. Analyses were undertaken as described by Zanewich and Rood [68] and others [16,69]. Briefly, tissue was extracted in aqueous methanol with internal standards of [^2^H_2_]-GA_1_, -GA_19_ and -GA_20_ (from L. Mander, Australian National University), filtered, and buffered (pH 9). Following *in vacuo* reduction, the aqueous phase was partitioned twice with water-saturated ether. The pH was then adjusted to 3 and the sample was further partitioned 3 times with ethyl acetate. Extracts were frozen, filtered to remove ice, and dried *in vacuo*. Residues were resuspended and transferred to glass microfiber discs and trace quantities of [^3^H]GA_1_ and [^3^H]GA_4_ were added for tracking. Samples were further purified using silica columns [68] and respective GA-containing fractions were separated using reversed-phase C_18_ high performance liquid chromatography. Grouped fractions containing the three GAs of interest were methylated and trimethylsilylated, and GAs were detected using gas chromatography-mass spectrometry (GC-MS) with selected ion monitoring. Quantities were calculated based in the paired ion abundances from the internal standard [^2^H_2_]GAs and endogenous forms [16,43].

### 4.5. Statistical Analyses

Analyses of variance (ANOVA) with SPSS v.19 (IBM, Armonk, NY, USA) provided the primary assessment for each growth measure with genotype and harvest day as fixed factors. These analyses used the data collected from harvest days 21, 35 and 42, which consistently displayed strong linear or log-linear patterns. Gibberellin content and concentrations were also analyzed by ANOVA, for all harvest days (three-factor: GA, genotype and day) or individual days (two-factor: GA and genotype).

## Figures and Tables

**Figure 1 plants-09-00139-f001:**
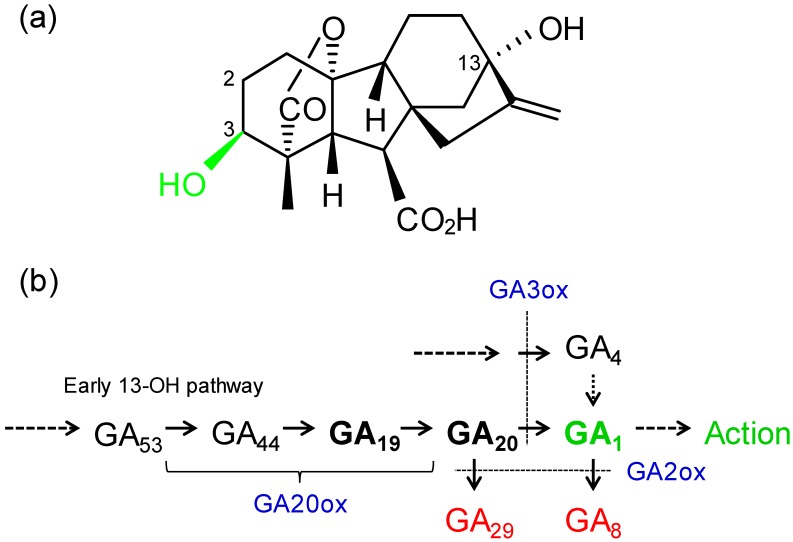
(**a**) Structure of gibberellin A_1_ (GA_1_), the bioeffector GA involved in the regulation of shoot growth and other processes in many higher plants. Numbers indicate positions of hydroxylations. (**b**) The early 13-hydroxylation GA biosynthetic pathway, including GAs and enzymes (blue; ox = oxidase) that were assessed in this study with *Brassica* or in the studies listed in Table 1 and Table 2. GA_19_ and GA_20_ are the primary GA_1_ precursors and GA_4_ provides an alternate precursor [16]. Following 2-hydroxylation, GA_29_ and GA_8_ (red) are biologically inactive.

**Figure 2 plants-09-00139-f002:**
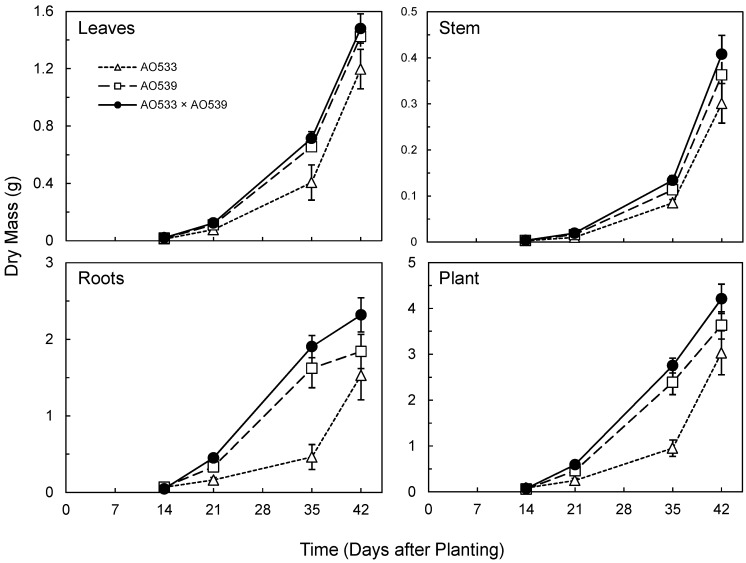
Dry mass of organs and whole plants of two *Brassica* inbreds and their F_1_ hybrid grown in greenhouse conditions (Means ± SE, n = 10).

**Figure 3 plants-09-00139-f003:**
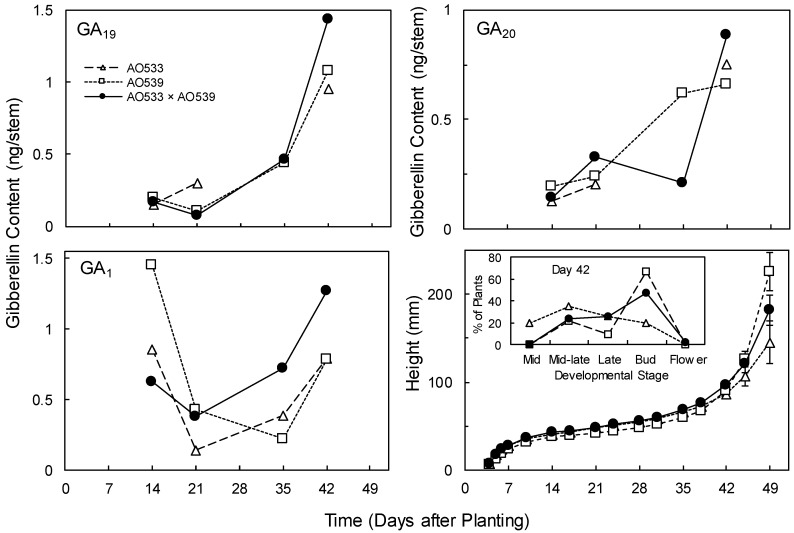
Contents of endogenous gibberellin A_1_ (GA_1_) and its precursors GA_19_ and GA_20_ in stems, and heights (Means ± SE provided for day 49, n = 10) of two *Brassica* inbreds and their F_1_ hybrid grown in greenhouse conditions. The inset plot displays the developmental stages at day 42, in accordance with Harper and Berkencamp [52].

**Table 1 plants-09-00139-t001:** Studies of gibberellins (GAs) and heterosis for growth in maize, or corn, *Zea mays* L., sequenced chronologically (GA_3_ = gibberellic acid).

Findings	Reference	Exogenous GA_3_	Endogenous GAs	GA metabolism	Gene expression	Other
Higher sensitivity of inbreds to exogenous GA_3_	[12]	X				
Higher levels of GA-like substances in shoot apices from a hybrid than parental inbreds	[17]		X			
Increased responsivity of inbreds, faster [^3^H]GA_20_ metabolism in a hybrid	[18]	X		X		
Faster [^3^H]GA_1_ metabolism in a heterotic hybrid	[19]			X		
Higher levels of GA_19_ and GA_1_ in hybrids than parental inbreds	[15]		X			
Increased GA_1_ and correspondence with amylase and seedling growth in a hybrid family	[20]		X			X
Correspondences between GA_3_ responsivity and GA content across inbreds and hybrids	[21]	X	X			
Heterosis persists with a GA dwarfing gene	[22]					X
Increased levels of ‘gibberellin-stimulated transcript 1 like protein’ in hybrids	[23]				X	
Increased expression of *GA 20-oxidase* in hybrids over parental inbreds, in multiple tissues	[24]				X	
Genome-wide associations support GAs as promoting brassinosteroids and heterosis for height	[25]					X
Induction of *GA 20-oxidase* promoted heterosis	[26]				X	

**Table 2 plants-09-00139-t002:** Studies of gibberellins (GAs) and heterosis in crop plants other than maize and trees, sequenced chronologically (GA_3_ = gibberellic acid).

Plant	Findings	Reference	Exogenous GA_3_	Endogenous GAs	GA metabolism	Gene expression	Other
Tomato	Greater growth promotion by GA_3_ of slow-growing inbreds	[13]	X				
Sorghum	Greater growth promotion by GA_3_ of shorter varieties	[27]	X				
Spruce	Increased responsivity to GA_4/7_ in slower growing F_1_ crosses	[28]	X				
Poplar	Higher levels of GA-like substances in bark scrapings from hybrids, GA_19_ and GA_1_ identified	[29]		X			
*Plantago*	Slower growing inbreds were more responsive to GA_3_; a GA inhibitor reduced growth especially in faster growing lines	[30]	X				X
*Plantago*	Decreased apparent GA, by immunoassay, in a slower growing inbred	[31]		X			
Poplar	Higher GA-like substances in bark scrapings from hybrids than parents	[32]		X			
Sorghum	Higher GA_1_ in fast-growing hybrids than parents, 2 triplets	[33]		X			
Sorghum	Increased [^3^H]GA_20_ metabolism in fast- growing hybrids	[34]			X		
Eggplant	Shorter genotypes were more responsive to GA_3_ for inbreds and hybrids	[35]	X				
Poplar	No increase in levels in subapical internodes of GA_44,20,29,1,8_ but harvests were after the growth interval	[36]		X			
Wheat	Higher GA_4_ in hybrids; increased expression of *GA 20-oxidase* and *GA 3-oxidase*	[37]		X		X	
Wheat	Heterosis for height correlated with expression of *GA 20-oxidase* and *GA 3-oxidase* variants	[38]				X	
Rice	Higher levels of GA_53,44,4,1_ but lower GA_20_, and increased expression of GA biosynthesis and action genes in hybrids	[39]		X		X	
*Arabidopsis*	A GA biosynthetic inhibitor blocks heterosis	[40]					X
Poplar	Across hybrids, growth positively correlated with GA_8_ but negatively correlated with GA_19_ and GA_20_	[41]		X			
*Brassica*	Shoot elongation correlated with GA_19,20,1_ but slight heterosis, primarily for increased root growth	This study		X			

**Table 3 plants-09-00139-t003:** Characteristics of two *Brassica* inbreds (AO533 and AO539) and their F_1_ hybrid, grown in greenhouse conditions (± SE, n = 10). Statistical comparisons including these results along with measures from plants harvested on days 21 and 35 are provided in Table 4.

Genotype	Seedling Emergence (%)		Foliar Characteristics at Day 42		
	Day 4	Day 7	Leaf Number	Leaf Size (cm^2^)	Leaf Area (cm^2^)
AO533	25.9	58.8	10.6 ± 0.4	32.5 ± 2.3	352.4 ± 37.5
AO539	41.2	89.4	12.3 ± 0.3	30.3 ± 0.9	371.2 ± 16.4
AO533 × AO539	71.8	95.3	12.8 ± 0.3	30.0 ± 1.7	384.6 ± 23.8

**Table 4 plants-09-00139-t004:** Analyses of covariance for growth characteristics of two *Brassica* inbreds (AO533 = “A3” and AO539 = “A9”) and their F_1_ hybrid grown in greenhouse conditions. Plants were harvested at 21, 35 and 42 days, and day provided the covariate. There were generally 99 plants producing *F*_(2,96)_, except for GA content (*F*_2,20_) and *, ** = *p* < 0.05, *p* < 0.01, respectively. For comparisons, genotypes are sequenced by decreasing values with least significant difference (LSD) pairwise comparisons: ‘~’ not significantly different, or greater (‘≥’ = *p* < 0.1; ‘>’ = *p* < 0.05, and ‘>>’ = *p* < 0.01). The potence ratio (PR, averages from 3 harvests for each value) provides a measure of dominance with values substantially exceeding 1 indicating overdominance, or heterosis (PR > 1.5 in red). The PR value for GAs is inflated due to similarity of the two parents for some GA measures.

Characteristic	Genotype F Value	Comparisons	Potence Ratio
Number of Leaves	22.10 **	F_1_ ~ A9 >> A3	1.57
Leaf Size	2.60	F_1_ ~ A9 > A3	1.35
Leaf Area	5.76 **	F_1_ ~ A9 > A3; F_1_ >> A3	1.81
Leaf Mass	8.53 **	F_1_ ~ A9 >> A3	1.58
Juvenile Height	5.48 **	F_1_ ~ A3 > A9; F_1_ >> A9	2.71
Stem Mass	4.45 *	F_1_ ~ A9 > A3; F_1_ >> A3	2.36
Root Mass	15.78 **	F_1_ *>* A9 >> A3	2.64
Total Plant Mass	15.76 **	F_1_ > A9 >> A3	2.23
Gibberellins (GAs; content/stem)	1.83	F_1_ ~ A9 ~ A3	15.8

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
