# Peer review of "Gibberellins and Heterosis in Crops and Trees: An Integrative Review and Preliminary Study with Brassica"

_plants, 2020, doi:10.3390/plants9020139_

Round 1

Reviewer 1 Report

The authors provide a good review of evidence in support of a theory of heterosis across a number of species and provide similar new evidence from an another taxa (Brassica), which also displays heterosis. The authors describe some interesting new findings related to heterosis of root mass, which raised questions about trait components of root mass heterosis. I feel like the manuscript failed to raise significant questions, gaps, or problems that would be solved or addressed by research conducted in this study, despite the somewhat lengthy Introduction justified as being a "Review". I also feel like the title and abstract lead one to believe that we are talking about heterosis in yield, but I am really sure if the theory to yield of the crops reviewed in this manuscript...except that the authors suggest that "Heterosis for root growth could contribute to yield stability and influence adaptation to drought and salinity." I would like to see the authors clarify how the theory and evidence discussed in the manuscript relate to yield. If this is not certain, then I think that the scope of inference needs to be clearly defined in the title and abstract.

Author Response

A single reply report is provided with combined responses for all 4 reviews - there were some overlapping aspects.

Reviewer 2 Report

The manuscript is an investigation on GAs involvement in heterosis, is more of review and a presentation of some preliminary data. Though the F1 progeny is highly diverse to be considered for this type of a study there is scientific value of the data presented.

The core contributor for heterosis seems to be root traits for F1 than the shoot traits that have been measured. If possible, a separate investigation on auxin quantification in parent and F1 line would add a lot of depth to this manuscript to see whether Auxin is the phyto hormone contributor for the heterosis of F1. However, the manuscript is well written but need further changes for publication.

Include the information of parent lines, which line was used as female and male line Table 4 legend - Define what is A9 and A3 (it is obvious but will make it east to read, >> has to be defined as 0.01 not 0.001 (does not match with the table values). The sample number information should be included in the legend. Overall this table should include pair wise comparison data as it will clearly show where the differences are (at least figures and p values both parent lines to F1. There is less value in here comparing the the two parent lines). The notations that has used to show the LSD is quite confusing and are not effective to gather information at a glance.  Figure 2, sample number information to be included. It could be seen that in some tables and graphs parent lines are denoted by A9/A3 and in other places with the full accession. Suggest this to be consistent throughout the manuscript. Discussion lacks the linking of current information to their findings. Need more explanation for the results obtained with the help of available literature. Material and methods should include the information of how the F1 was produced, which parent was male and female line.

Author Response

(The authors gave the same response as above.)

Reviewer 3 Report

The manuscript titled “Gibberellins …Brassica”  attempts to address the basic principles of stem growth with relevance to trees and crops through heterosis, giving a very interesting point of view on the role of Gibberellins. This is the subject of great interest to readers across diverse fields in plant biology. The article provides general background and historical context for readers by revisiting previous studies on GA mediated heterosis. The authors also present elements of the GA biosynthesis pathways and improved hormone quantification techniques of GC-MS while emphasizing common pitfalls in tissue sampling for quantitation. The manuscript also includes the studies on heterosis in Brassica, in which modest increase in growth and bioactive GA were found in F1 hybrids. Finally, the authors settle their question on ambiguous position of GA in mediating heterosis. This is anticipated because other hormones may operate independently or in association with Gibberellins to promote vigorous growth. Overall, the manuscript is written well; nevertheless, I suggest authors to integrate recent views in the field to provide big picture information to the readers.   

Line 114: “greater dilution by the larger stem of hybrids:” The authors make a good point to explain the contradictions across studies. I would urge authors to add more caveats on those lines and provide potential solutions to circumvent such problems.

Line 114: It may be that such dilutions are indeed necessary for rapid increment in shoot growth? –Which is unknown so far. As mentioned above, various phytohormone pathways such as auxin, brassinosterioids, controlling growth may interact (Wang et al. Auxin pathway control hybrid vigor PNAS April 25, 2017 114 (17) E3555-E3562; Gibberellins Promote Brassinosteroids Action and Both Increase Heterosis for Plant Height in Maize (Zea mays L.) Hu et al., Front. Plant Sci., 2017   https://doi.org/10.3389/fpls.2017.01039). Therefore, given such complex interactions, the linearity in GA levels may not be anticipated.

Line 114: Growth could be resulting from both mechanisms involving-  increase in cell number and cell expansion (larger size). For example, hypocotyl growth involves interactions between cell division and cell expansion. These cases would also result in departure in expectations when quantifying GA. Additionally, the tissue type used across studies may vary- for instance, hypocotyls vs elongating stem with marked internodes.

Table 2:  A) Here, the authors include studies involving quantification of GA hormone per se and biosynthetic/catabolic gene expressions datasets (Rice, wheat).  Generally, the gene expression variations arise due to allelic complementation, presence/absence of genomic region/chromosomes, and underlying complex molecular interactions. Likewise, protein structure complementation as described in the classic case of adh1 locus studied by Schwartz and Laughner (Science, 166, no. 3905, pp. 626–627, yr 1969).  I would suggest the authors to allocate a separate paragraph describing genetic mechanism in F1 hybrids when referring to these datasets. The gene expression changes in F1s might not be fully justified by dominance or overdominance gene actions.

Table 2; B) The authors present cases of polyploidys such as Wheat. Polyploidy present a complicated situation because inbreeding resulting in homozygosity needs to be achieved for each of the contributing sub-genomes. Furthermore, there would be interactions between sub-genomes from contributing parents and would potentially involve genetic interactions due to dosage of the genes, complementing transcription factors etc. that would eventually affect the quantification of gene expression. I would suggest the authors to add a comment when drawing inferences from gene expression studies in wheat (Ref. 37,38).

Gibberellins and Heterosis in Brassica: The study involves two parental inbred AO533 and AO539. How many generations of selfing in parents were done prior to crossing for F1 seeds so that inbreeding depression/ homozygosity was achieved?  Would this be one of those factors underlying moderate heterosis observed in this study?  What is the source of parental lines or genetic pool? Are the parental lines selected from an elite breeding program or landraces or wild populations? Inbreeds from breeding program may have been selected to decrease the recessive characters or deleterious alleles, hence increase the likelihoods of modest heterosis. Please furnish such details and address relevant topics.

Line 138 and section 3.2 : “pattern of dominance or slight overdominance”-Here, the authors describe the F1 performances in terms of dominance or overdominance gene models.  Overdominance and heterosis are also used interchangeably in the manuscript.  I suppose that dominance and overdominance gene model explaining the basis of heterosis fits well with the traits controlled by single gene- similar to dwarf mutants (cited in Line 44-45) in crops. Nevertheless, dominance/overdominance model laid foundation for heterosis.  Here, leaves, root, whole plant biomass are sampled in Brassica rapa, in which dominance or overdominance gene actions appear overgeneralized. Furthermore, recent studies implicated altered circadian rhythms and improved photosynthesis in mediating heterosis in Arabidopsis ( Ng et al., Plant Cell. 2014;26(6):2430–40) and maize (Ko et al., 2016 PLoS Genet.  12(7):e1006197. doi: 10.1371/journal.pgen.1006197 ). Thus, likely involve multiple genes or polygenic control and this aspect has been indicated in Line 238-239. I would suggest that authors to dedicate a separate paragraph to emphasize these gene action models (see comment above Table 2A) and integrate those recent studies providing insights into physiological basis of heterosis.

Author Response

(The authors gave the same response as above.)

Reviewer 4 Report

The manuscript entitled: "Gibberellins and Heterosis in Crops and Trees: An integrative Review and Preliminary Study with Brassica" written by Zanewich and Rood, presents in two parts an overview of the potential involvement of GA content for the heterosis effects in crops and trees, followed by a preliminary analysis of in Brassica of two lines AO533 and AO539, and their F1 hybrid, for selective traits with an attempt to link some of the heterosis effect to differences in the GA levels.

The title fully matches the text and is informative of the presented data. The text is nicely written and reads well (as far as my English proficiency allows me to proof it). The manuscript is interesting and presents an interesting research direction.

For the first part, there is not much to comment on, as the text is clear on what the authors wish to say. It is well documented.

Only, I got confused in part 1.3 when the levels of GAs are presented and compared to growth aspects of the hybrids. I understand from part 1.2 that hybrids are more responsive to exogenous GA applications, that would imply they do so because they would have a lower content in GA, and this is stated as a hypothesis at the beginning of part 1.3. However, this is contradicted in the following text still in part 1.3 where it is written that hybrids would have higher levels of endogenous GAs, which the authors wrote it support the hypothesis. In my opinion, it is opposite to the stated hypothesis. Can the authors clarify this point?

The second part presents preliminary data of heterosis analysis in Brassica (B. napus, B. rapa? Please specify in the main text) with an attempt to correlate it with GA levels. However, the two genotypes selected have only slight differences (although significant), in the analysed traits. The F1 hybrid behaves similarly in most of the case to the strongest parents. GA1 levels are the most interesting: lower in the F1 at young stages (may explain the fast growth), while higher in older stages. This last has been explained by advancing into the reproductive stages. As mentioned it is very difficult to state on any correlations at this point. Measurements in roots, detailed root architecture analysis and expression of key GA metabolic genes would be more than helpful in the interpretation of these data in a follow-up study. 

Data are well-interpreted and discussed. As mentioned, better-selected genotypes for specific traits may help these analyses. 

Material and Methods are complete, statistical analysis is strong (as far as my knowledge allows me to judge).

Author Response

(The authors gave the same response as above.)

Round 2

Reviewer 1 Report

The revisions are well done, the authors have addressed my comments.

Author Response

Please see attachment - we provide a single response report for all 3 reviews.

Reviewer 2 Report

Thorough spell check is recommended

Author Response

Please see the attachment that we posted for Review 1. We provide a single response report for the three reviews.

Reviewer 3 Report

The authors have attempted to address a few of the previous comments. It appears that they would like to remain focused on the physiological basis of heterosis via GA. However, the authors are not enthusiastic to reveal the pedigree  and source of parental inbreds used in this study. The authors mention in their response letter that parental lines were chosen based on the recommendation from the breeding program, which conflicts with the intent of the studying GA based heterosis. The identity of  lines must be associated with germplasm source or breeding program so that readers who are interested in exploring further phenotypic/physiological basis are careful not to replicate their experiments using similar parents or gene pool or selfed generations. Furnishing such information would also be best in the interest of fulfilling material and methodology.

Author Response

(The authors gave the same response as above.)
